# RadViz++: Improvements on Radial-Based Visualizations

**Lucas de Carvalho Pagliosa** *  **and Alexandru C. Telea** *

Faculty of Science and Engineering, University of Groningen, Nijenborgh 4, 9747 AG Groningen, The Netherlands

**\*** Correspondence: l.de.carvalho.pagliosa@rug.nl (L.d.C.P.); a.c.telea@rug.nl (A.C.T.)

**Abstract:** RadViz is one of the few methods in Visual Analytics able to project high-dimensional data and explain formed structures in terms of data variables. However, RadViz methods have several limitations in terms of scalability in the number of variables, ambiguities created in the projection by the placement of variables along the circular design space, and ability to segregate similar instances into visual clusters. To address these limitations, we propose RadViz++, a set of techniques for interactive exploration of high-dimensional data using a RadViz-type metaphor. We demonstrate the added value of our method by comparing it with existing high-dimensional visualization methods, and also by analyzing a complex real-world dataset having over a hundred variables.

**Keywords:** radial-based visualizations; data analysis; edge bundling; visual scalability

## 1. Introduction

Methods to study multidimensional datasets are a core topic in Visual Analytics [1]. Analyses supported by such methods can be divided into three classes: (i) data-to-data; (ii) data-to-variable; and (iii) variable-to-variable. The first type of analysis generally consists of Dimensionality Reduction (DR) methods that project data into a low-dimensional space to visually search for clusters and patterns [2]. While aiming to preserve data-to-data relationships, DR methods by themselves do not explain the *variable* space or, e.g., which variables impact the projection the most. Doing this requires additional visual metaphors [3–5]. On the other hand, methods like Parallel Coordinate Plots [6] and Scatterplot Matrices [7] help to perform data-to-variable analyses, but problems such as visual clutter and limited usability tend to occur when tens of variables or more are analyzed, hindering data-to-data correlation. Lastly, histograms and box-plot-based metaphors [8] can show distributions and similarities of variables, but also are limited for high-dimensional data as a large visual space is required to fairly compare several variables.

Overall, most high-dimensional visualization methods are mainly designed to tackle one (two, at most) type of analysis. Conversely, the radial visualization (RadViz), originally proposed by Hoffman et al. [9], is one of the most popular techniques [10] that perform all three types (i–iii) simultaneously. In this metaphor, each variable is mapped as an anchor along the circle such that data instances (represented as 2D points) are pulled towards them according to their respective variable values. In this context, while data information can be extract by analyzing the formation of clusters and outliers inside the circle (data-to-data), those patterns can be explained by the proximity of data points to the anchors (data-to-variable). In addition, variables are correlated according to their distance or the order they appear in the circle (variable-to-variable).

Despite benefits, however, RadViz-class methods can also lead to misconceptions and clutter when different instances are mapped into the same visual location. These so-called *ambiguities* [10,11], the dependency to anchors positioning, and the limited space in the circle contribute to a generally

lower ability to separate same-data-sample clusters than e.g., DR methods [2]. In this context, methods in the literature (Section 2.2) proposed to optimize how anchors are ordered in the circle, but solutions are still restricted for a relatively small number (few tens) of variables. In summary, we identify the following possible improvements for RadViz-class visualizations:

R1    Be scalable in both the number of variables and instances;
R2    Decrease and/or explain visual ambiguities they create in data-to-variable analyses;
R3    Show unambiguously variable relations to support variable-to-variable analyses;
R4    Separate data clusters well to support data-to-data analyses.

Based on those requirements we propose RadViz++ (the source code can be found in https://github.com/pagliosa/radviz-plus-plus), a novel RadViz-class technique to support tasks (i-iiii) while better satisfying R1-R4. We order variables along the circle following the hierarchical clustering based on variable correlations, and draw clusters compactly using an icicle-plot metaphor [12]. Scalability is addressed by allowing users to interactively aggregate and/or filter out variables while exploring how this changes data-to-data insights. We add histograms over each icicle-plot cell to show its respective variable distribution. Besides showing this, one can select histograms bins to filter data based on ranges of multiple variables. Conversely, we use a brushing-and-linking metaphor to select data points and explain them by their respective variable bins, thereby decreasing ambiguity issues. We use an edge-bundling technique [13] to show strongly correlated variable anchors, thereby clarifying variable-to variable relations. Finally, we allow smoothly animating between the RadViz scatterplot and a classical DR scatterplot to let users link cluster (best shown by the latter) by variables that explain them (best shown by the former).

We structure this paper as follows. Section 2 presents concepts and problems related to RadViz-class methods and discusses related work. We introduce RadViz++ in Section 3. Section 4 shows RadViz++ using synthetic and real-world datasets. Section 5 discusses how our method helps with requirements R1–R4. Section 6 concludes the paper and outlines future work directions.

## 2. Related Work

We firstly describe the fundamental concepts and problems of RadViz-class visualizations. Next, we present how state-of-the-art methods tackled those issues, and where they can be improved.

### 2.1. Concepts and Background

Following the nomenclature of RadViz Deluxe [14], consider a multidimensional dataset, represented in matricial form as

$$X = \begin{bmatrix} x_{11} & x_{12} & \cdots & x_{1n} \\ x_{2n} & x_{22} & \vdots & \vdots \\ \ddots & \ddots & \cdots & \vdots \\ x_{m1} & x_{m2} & \cdots & x_{mn} \end{bmatrix}, \tag{1}$$

where $m$ and $n$ are the number of instances (also called samples or observations) and variables (also called attributes, dimensions, or features), respectively. In this context, a RadViz-class visualization [15] maps the variables $V_1, \ldots, V_n$ (columns of $X$) to so-called *anchors* $v_1, \ldots, v_n$ on the circle boundary (with radius $r$) as

$$v_j = \left( r \cos \frac{(j-1)2\pi}{n}, r \sin \frac{(j-1)2\pi}{n} \right), \tag{2}$$

so that instances $D_i$ (rows of $X$) are represented by points $P_i$ according to

$$P_i = \sum_{j}^{n} \frac{x_{ij}}{\sum_{j}^{n} x_{ij}} v_j. \tag{3}$$

In this context, $P_i$ is 'pulled' towards the anchors $v_j$ proportionally to the its positive value $x_{ij}$ (Figure 1). If we use the same logic, $P_i$ should be repelled by the same force if negative values were allowed. Yet, this would be misleading, since repelling a point from an anchor $v_j$ inevitably pushes it to some other anchor along the circle boundary, opposite of $v_j$. Separately, normalization of Equation (3) is needed to ensure all points are mapped inside the circle, which is not guaranteed when negative $V_j$ values are involved. Therefore, negative values are usually handled by either normalizing $V_j$ to $[0, 1]$ or taking their absolute values. However, properly normalizing is hard as the proportionality of variables over instances can be lost.

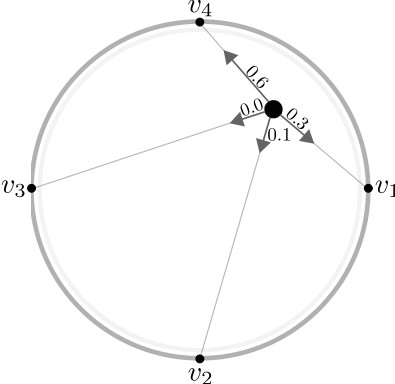

**Figure 1.** An instance is pulled towards the anchors proportionally to its normalized variable values.

Nonetheless, visual ambiguities are still a problem even after the above steps, see instances $D_1$, $D_2$, and $D_3$ in Table 1. As we can see, normalizing the variables of those three instances (to remove negative values) maps $D_1$ and $D_2$ to the same point. Similarly, $D_2$ and $D_3$ get overlapped if absolute variable values are used.

**Table 1.** Different inconsistencies that can occur in RadViz. Instances $D_1$, $D_2$, $D_3$ get mapped to the same point after procedures to avoid negative numbers. On another hand, instances $D_4$, $D_5$, $D_6$ show the typical sensitivity to anchor positioning in RadViz designs.

| Instance | $V_1$ | $V_2$ | $V_3$ | $V_4$ |
|:---:|:---:|:---:|:---:|:---:|
| $D_1$ | 0 | $-20$ | $-60$ | $-60$ |
| $D_2$ | 20 | 40 | 80 | 80 |
| $D_3$ | 2 | 4 | 8 | 8 |
| $D_4$ | 0 | 0 | 0 | 0 |
| $D_5$ | 20 | 1 | 20 | 1 |
| $D_6$ | 100 | 5 | 100 | 5 |

Thus, the most common form of ambiguity in RadViz-class methods occurs when points are 'pushed' to the circle center (even when only positive values are used), either because instances have equal variable values ($D_4$) or when a subset of anchors is placed so that their forces cancel each other ($D_5$, $D_6$). To alleviate this, several methods try to optimize anchor placement and how points are attracted to them (Section 2.2). Yet, inconsistencies caused by the anchors placement will eventually occur, especially when the number of variables increases. This is due to the inherent limits of the circular space along which anchors are placed. Due to these limits, we need ways to *disambiguate* different instances that get mapped at similar locations.

## 2.2. Related Methods

As cyclic ordering is an NP-complete problem [16], several heuristics were suggested to optimize circular anchor placement to decrease ambiguities. For example, to separate different instances that get overlapped in a classical RadViz plot, Nováková and Štěpánková [17] propose a 3D RadViz design where instances are drawn into the $xy$ plane via Equations (2) and (3), while their instance norm is mapped to the $z$ axis. This addresses R2, as instances like D5 and D6 (Table 1) can be distinguished by their heights while viewing the 3D layout from different viewpoints. Yet, this does not tackle scalability (R1) nor support deeper variable-to-variable analysis (R3). Moreover, finding a suitable 3D viewpoint can be hard.

The Mean Shift (MS) method [18] partitions each variable into several new variables according to its probability distribution function. The procedure repeats for each variable as follows. First, the distribution of $V_j$ is discretized into a histogram of $p$ bins, whose density values are interpreted as 1D points. These $p$ points are clustered by a Gaussian-based technique that maps points to the centroid of their neighbors (all points inside the kernel bandwidth). After all bins converge to a centroid, $V_j$ is partitioned into new variables according to each centroid interval. Moreover, $V_j$ is removed from the visualization and the new variables added. Finally, all variables are placed along the circle to optimize the Dunn index [19], exhaustively calculated for all possible combinations of anchor positions. This method can be seen as an extension of Vectorized RadViz (VRV) [20], proposed to analyze categorical data. Both MS and VRV aim to decrease ambiguities (R2), higfight interval-basis similarities among variables (R3), and aim to better cluster the data (R4). Yet, both methods have an even lower dimension-scalability (R1) than classical RadViz as they can accommodate fewer variables while keeping visual clarity after the partitions.

Ono et al. [21] propose Concentric RadViz (CRV), an interactive tool for multitask classification. Variables are clustered according to their tasks into concentric circles following [22]. Sigmoid normalization is applied to ensure all points remain inside the circle, even when nested anchors are aligned. Users can rotate anchors in any direction and at any level to analyze the formation of patterns and correlate instances over multiple tasks. CRV can also be seen as an extension of Star Coordinates (SC) [23], where users can rotate and scale anchor positions at will, starting from an initial equally-distributed anchor placement along a single circle. Both CRV and SC handle well datasets with a few tens of variables. For more variables, the interactive search for a good anchor placement become hard as there is no visual cue to guide users during this search (limited R1 support). Moreover, both methods eventually lead to clutter even when multiple circles are used (problems with R1 and R2). However, structures (clusters) are potentially better represented after interactions (R4). Finally, R3 is partially addressed as users can correlate variables not only by their distances but also by how they are aligned in the circular hierarchy.

Also an extension of SC, iStar [24] is an interactive tool that, besides allowing traditional scale/rotate operations of the variable axes, also supports the union and separation of axis anchors at will, readjusting data points in real time. To support R1 for large numbers of variables, these can be clustered automatically by the $k$-means algorithm [25] based on their variance, bidimensional PCA coordinates, or centroids of classes (when class labels are present). Next, given the matrix $M$ of variable pairwise similarities, a graph is created where nodes are anchors and edges have lengths given by the pairwise similarities $M_{ij}$. The ordering of anchors around the circle is given by the optimal closing path connecting all nodes, computed using a Genetic Algorithm [26]. The distance between adjacent anchors is given by their edge length, thus, similarity. Given their design, iStar axes are related to biplot axes, well known in information visualization [27–29]. iStar supports R1 very well, showing dataset examples of hundreds up to thousands of instances and variables. Variable-to-variable analyses are also well supported by the proposed clustering (R3). However, setting the number of $k$ clusters in $k$-means is no trivial task—this works well only if the user has *beforehand* a good idea of how many groups-of-variables he/she would like to simplify the data into, which similarity metric to use for the variables, and if the variables are indeed distributed this way in the data. Similarly, despite that

iStar allows users to freely joint and split variable groups, there is no visual cue to *guide* this process, besides the formation of point-group structures in the plot *after* the respective user action was done. iStar does not tackle R2 as there is no visual metaphor provided to explain ambiguous points. Finally, iStar can achieve quite good cluster separation, as demonstrated on many datasets (R4). However, this requires careful user intervention in terms of selecting $k$, as well as manual anchor arrangement, grouping, and filtering.

From a different perspective, Rubio-Sánchez et al. [30] proposed to use the user-defined anchor positions from SC to minimize $\left\| PA^T - X \right\|_F^2$, where $A$ is the $n \times 2$ matrix composed of 2D anchor vectors, $P$ is the $m \times 2$ matrix containing the 2D coordinates of the scatterplot points, and $\| \cdot \|_F^2$ denotes the Frobenius norm. The authors also apply a kernel function to $A$ to make its columns mutually orthonormal, which provides "a more faithful representation of the data since it avoids introducing distortions, and enhances preserving relative distances between samples". The above minimization improves R4 and partially fulfills R2, as there are no metaphors to explain data-to-variable analysis ambiguities. Finally, the method does not extend variable-to-variable analysis (R3) with new solutions, nor does it explicitly address dealing with large numbers of variables (R1).

Recently, the RadViz Deluxe (RVD) [14] method aims to improve the quality of all analysis types (i) to (iii). RVD proposes different methods to reduce errors of the low-dimensional representation, namely variable-to-variable, data-to-variable and data-to-data errors, in this order, as follows. First, anchor placement along the circle is computed by an approximate Hamilton Cycle solution [31], so that distances between adjacent anchors reflect their pairwise correlations. Secondly, the data-to-variable error is decreased by a series of iterative geometrical operations. Finally, the data-to-data error is reduced by a spring system similar to [32], where an instance $D_i$ is attracted (respectively repelled) to $D_j$ if their distance in $n$D space is smaller (respectively greater) than in the 2D visual space. Despite improvements regarding R2 and R4, RVD still lacks solutions for R1 (scalability) and R3 (variable-to-variable analysis). Moreover, RVD reduces errors following a fixed pipeline. Hence, it is likely that after changing the visualization to decrease one error (e.g., data-to-data), other errors increase (e.g., data-to-variable and variable-to-variable). Finally, let us recall that a main proposal of RadViz is to explain the projected data *and* their variables. Consider Figure 2a, generated by RadViz. Here, anchors correctly describe (explain) data points. For instance, the black outlier point, close to anchor $v_1$, has variation only in variable $V_1$. This explanation is partially lost by the RVD layout corrections, as data points are not strictly represented by anchors anymore. Consider Figure 2b, generated by RVD. Point clusters are indeed better separated now. However, anchors cannot be used to reliably explain the points. For instance, the black outlier moved towards the center, what could give the wrong impression that it may also have positive values in $V_2$, $V_3$ or $V_4$. Figure 2c shows the difference between the first two figures.

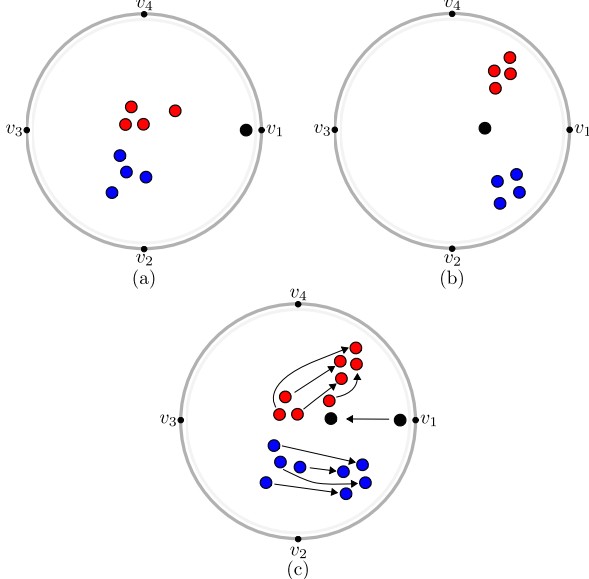

**Figure 2.** (**a**) RadViz representation of a simple dataset showing clusters (red and blue) and one outlier (black). (**b**) RadViz Deluxe layout of the same data showing better cluster separation but poorer explanation of the outlier. (**c**) Differences highlighted between (**a**,**b**).

## 3. RadViz++ Proposal

To address the requirements listed in Section 1 and to alleviate the observed limitations of current methods, we propose RadViz++, a novel radial-based visualization for high-dimensional data. RadViz++ allows users to interactively aggregate, separate, and filter variables, and see in real time how this impacts the layout on a data-to-data, variable-to-variable and data-to-variable basis. We next introduce and explain the features of RadViz++ and outline how they address R1–R4 and also improve upon related RadViz-class methods. We use as running example the well-known Segmentation dataset [33–35], which has $m = 2100$ instances, $n = 18$ variables, and 6 instance classes. For conciseness, the variable names are next referred to as $V_1, V_2, \ldots, V_{18}$. Instances are randomly-chosen $3 \times 3$ pixel blocks from seven manually-segmented outdoor images. Variables are statistical image attributes, such as color mean, standard deviation, and horizontal/vertical contrast, often used in image classification. The class attribute denotes the image type. Visual analysis tools use this dataset to discover how specific sets of variables and/or variable ranges can explain the similarity of groups of points, see e.g., [4,36]. In turn, this can help designing better feature-engineering-based classifiers for such data.

### 3.1. Anchor Placement

As a baseline, we show the results of the original RadViz method on the Segmentation dataset (Figure 3a). Here, anchors $v_i$ are placed anticlockwise along the circle in the order their variables $V_i$ appear in the dataset. Here and next, scatterplot points are color-coded on their class label. As visible, no clear cluster separation can be seen. Yet, we know that such a separation does exist [4,33,34,36].

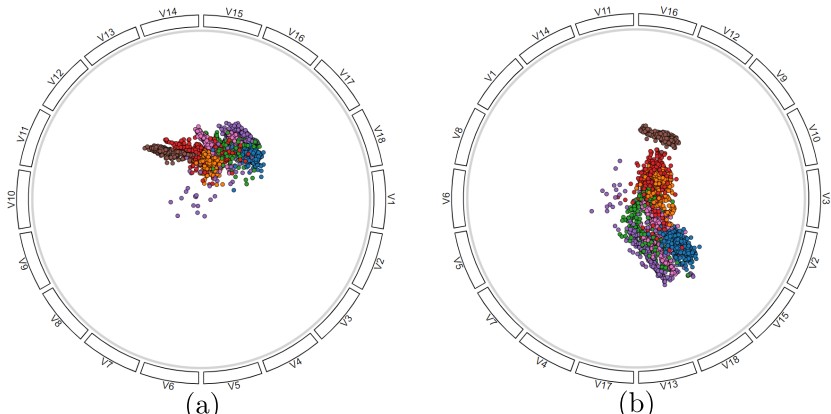

**Figure 3.** (**a**) RadViz with no variables ordering. (**b**) in RadViz++, anchors are rearranged in the circle according to their correlation coefficient. In our implementation, anchors are depicted by cells with the corresponding variable names above them, and points are colored based on their classes.

To see this separation, a better approach is to order anchors based on the similarity of their variables. Among many ways to compute this similarity, known in the time-series literature, e.g., AMI [37], DTW [38], ARIMA [39], we choose the Pearson correlation coefficient [40], similarly to RVD, given its simplicity. Hence, the similarity of $V_i$ with $V_j$ is given by

$$\rho(V_i, V_j) = \frac{\mathrm{Cov}(V_i, V_j)}{\sqrt{(\mathrm{Var}(V_i) \times \mathrm{Var}(V_j))}}. \tag{4}$$

To obtain a similarity metric, we normalize $\rho$ to $[1, 0]$. To place anchors, we next compute the all-variable-pairs distance matrix $A_{ij} = \rho(V_i, V_j), 1 \leq i \leq n, 1 \leq j \leq n$, and next cluster the variables $V_i$ via average-linkage Agglomerative Hierarchical Clustering (AHC) [41]. Figure 4a shows the cluster dendrogram produced for our running dataset. In contrast to RVD, we now arrange anchors around the circle in the order that leaves appear in the dendrogram (Figure 3b). As visible, clusters already get better separated than in the original RadViz layout (Figure 3a). In addition, we show in Section 3.4.1 how we can use the hierarchy to address scalability for many variables (R1) and also a better cluster separation and explanation (R4).

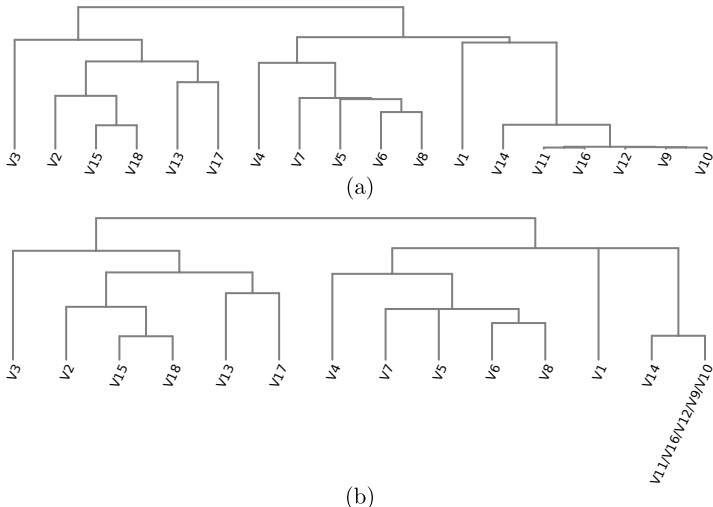

**Figure 4.** (**a**) Dendrogram built from variable correlation (Section 3.1). (**b**) Simplified dendrogram (Section 3.2.1).

Despite this approach now leads to two clusters instead of one, it is still not enough to achieve an optimal representation. Regarding the distance among anchors, it is worth to mention that we *accept* the fact that, as dimensionality increases, it becomes more difficult to place all variables well separated in the circle according to their similarities. Therefore (and also in contrast to RadViz Deluxe), we make all anchors equally sized so neighbors have the same distance among themselves so that we can fit more variables in the same amount of visual space without clutter.

### 3.2. Variable-to-Variable Analysis

Atop of the hierarchical variable placement described in Section 3.1, we propose two visual metaphors to help variable-to-variable analysis in different levels, as follows.

### 3.2.1. Variable Hierarchy

We draw the variable dendrogram using a circular icicle-plot metaphor where all leaves are aligned at the same level. A similar layout for hierarchical data was used by Holten [13] for displaying different data types (software containment) and in a different context (program comprehension). As a key difference, icicle-plot cells in our case are groups of similar (correlated) variables, and not data instances. Cell colors indicate variable similarity using a blue (similar) to green to red (dissimilar) ordered colormap. Labels atop cells show the variables these aggregate.

In this context, depicting the full dendrogram produced by AHC typically demands too much space in the visual plot, since each binary clustering event creates a new level. Figure 5a shows the resulting icicle plot for the dendrogram in Figure 4a. Hence, we simplify the dendrogram by aggregating variables (by summing their respective values) having parents that are more similar than $\delta = 10\%$ of the root cluster diameter. A similar approach was used in a different context by Carlsson and Mémoli [42]. Figure 4b shows the simplified dendrogram for our running example dataset. Thus, increasing this value yields a simpler dendrogram which needs less visual space, but shows less details on how variables relate to each other. Conversely, decreasing this value yields more details on the similarity values of variable pairs, but requires more visual space. Figure 5b shows the visualization of this simplified dendrogram.

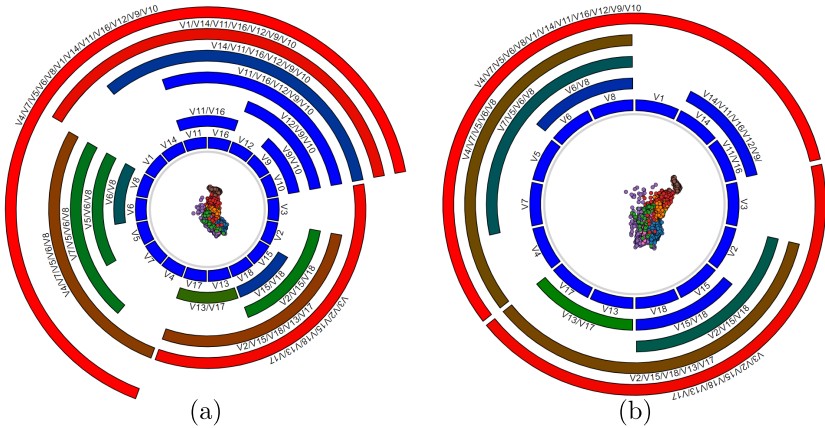

(a)        (b)

**Figure 5.** (**a**) Circular icicle plot showing the full dendrogram ($\delta = 0\%$). (**b**) Plot of the simplified dendrogram ($\delta = 10\%$) leading to a more compact layout.

### 3.2.2. Similarity Disambiguation

The icicle plot described above addresses the task of finding *groups* of similar variables, as children of the same node in the plot. However, the plot does not (easily) support the task of finding how similar a group of variables is to *other groups*. To see this, one needs to carefully study the entire icicle-plot hierarchy, including comparing the colors of multiple nodes. To support this task, we adapt the Hierarchical Edge Bundling (HEB) [13] technique as follows. We consider a graph $G$ where each node

is an anchor $v_i$, and each edge is the similarity $\rho(V_i, V_j)$ between variables $V_i$ and $V_j$. We then construct the HEB bundling of $G$, using as hierarchy the one given by the (simplified) dendrogram, to draw it such each edge has $\rho$ encoded into its opacity. Figure 6 shows the result. The less correlated two variables are, the more transparent and closer to the circle center will be its bundled edge. Conversely, strongly correlated variables will have dark (opaque) and far-from-center bundled edges. Bundles thus show groups of variables which are similar to each other.

Bundling serves an additional disambiguation task. As explained in Section 3.1, for a sufficiently large variable count $n$, it becomes hard, and in the limit impossible, to assign positions for the anchors $v_i$ along a circle so that distances along the circle *accurately* reflect high-dimensional similarities of the variables $V_i$, no matter which anchor placement strategy we use. This is the well-known distance preservation problem in dimensionality reduction when going from $n$ dimensions to a single one. Moreover, the *circular* nature of the RadViz design will place variables which are at opposite ends in the (simplified) cluster tree ($V_3$ and ($V_{11}, V_{16}$) in Figure 4b) next to each other along the circle (Figure 6). The same happens for variables $V_4$ and $V_{17}$. Without any other visual cue, one may think that these are very similar variables. The HEB bundles solve this. As no dark bundle connects these cells in Figure 6, they do not contain similar variables.

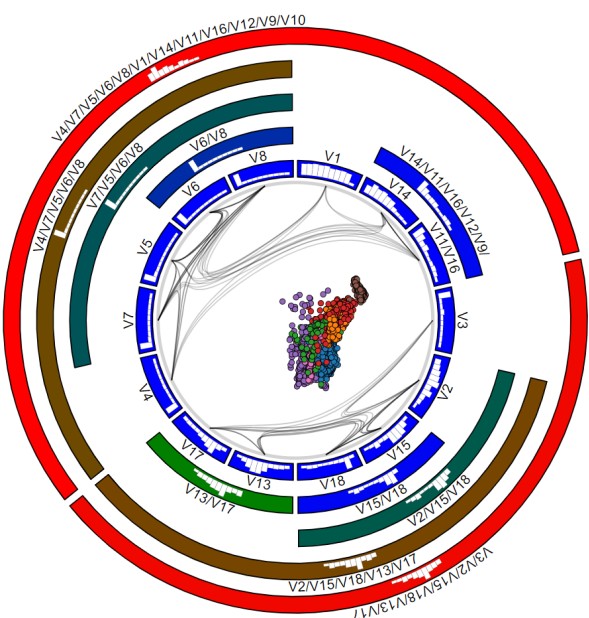

**Figure 6.** HEB bundles and variable histograms in RadViz++.

### 3.3. Analyzing Variable Values

The mechanisms discussed so far show us which variables are similar, but they do not explain in detail why. Moreover, one is often interested in explaining the similarity of instances not only in terms of entire variables, but *ranges* of values thereof. To support such tasks, we plot histograms over each icicle-plot cell to show the respective variable distributions. By default, we use $h_{def} = 10$ histogram bins. However, icicle-plot cells can have widely different sizes, depending on the dendrogram clustering and total number of variables. For over a few tens of variables, some cells become too small to display 10-bar histograms. Varying the visual width of a histogram bar on the cell size is not a good idea, as it makes comparing histograms in different-width cells hard. Hence, we fix the width of a histogram bar to $w_h$, set in practice to 5 pixels, and use $h = \min(h_{def}, w_c/w_h)$ histogram bins for a cell of width $w_c$. This way, smaller cells will display fewer-bin histograms (see e.g., Figure 6).

Besides seeing the value distributions of each variable $V_i$, histograms have two other uses. First, they allow comparing different variables. For instance, in Figure 6, we see that $V_5$, $V_6$, $V_7$, and $V_8$ are strongly correlated (since linked by bundles and children of a grandparent node colored dark-green),

*and* they also have very similar distributions, with mostly small values. In contrast, nodes $V_{15}$ and $V_{18}$ show a similar correlation (same dark blue color), but quite different distributions. Secondly, histogram bars can be interactively clicked to select points whose values belong to the selected bins. This serves a explaining which variable *ranges* are responsible for certain patterns in the scatterplot, but also b de-clutter scatterplot areas where multiple points are plotted atop each other.

### 3.4. Scalability and Level-of-Detail

We address both these issues by aggregating and filtering variables and data points, as follows.

#### 3.4.1. Aggregating Variables

The key purpose of the icicle plot is to show how the data can be explained in terms of *groups* of similar variables. In the case when the user decides that all child variables of a parent node in this plot can be seen as a single one, displaying all of them makes the visualization unnecessarily verbose. Clicking such a parent node aggregates all its child variables, replacing them with the centroid value of the respective AHC cluster, and regenerates the visualization. Figure 7a shows this after we aggregate variables $V_2$, $V_{13}$, $V_{15}$, $V_{17}$, $V_{18}$ (large brown cluster, Figure 6 bottom); variables $V_6$, $V_8$ (Figure 6, top-left blue cluster); and variables $V_9$, $V_{10}$, $V_{11}$, $V_{12}$, $V_{14}$, $V_{16}$ (Figure 6, top-right blue cluster). The former aggregation, however, leads to more overlap in the scatterplot—hence, this simplification level may be too strong to allow us to correctly interpret the data. To fix this, we do one step backward by clicking on the large brown cluster in Figure 7a to split it into its direct children. The result (Figure 7b) shows a very similar scatterplot to the original, unaggregated, one (Figure 6). This plot is obtained by using only *nine* variables (either original ones or aggregations) as compared to the original 18. Hence, we obtain a 50% dimensionality reduction with little loss of the data structure.

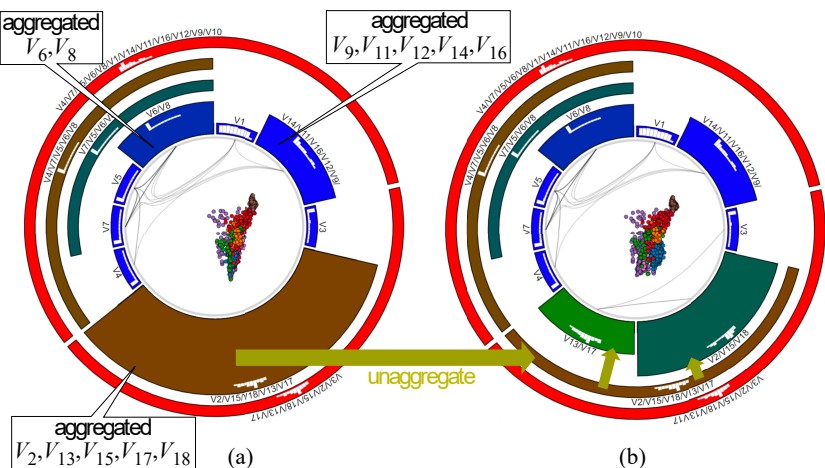

**Figure 7.** (**a**) Aggregation of several variables. (**b**) Refining the aggregation for the bottom (brown) cluster.

#### 3.4.2. Variable Filtering

While useful, variable aggregation has the problem that it actually *synthesizes* new variables from existing ones. This is not always desirable, e.g., when certain variables do not logically make sense to be averaged together. Conversely, there are cases when we want to completely eliminate, or *filter* away, certain variables, e.g., which we recognize as not useful for the analysis. By clicking icicle-plot cells the user can also filter away desired variables, after which the remaining space is reallocated to the remaining variables. Figure 8 illustrates this. First, we decide that only variables in the colored cells are to remain after filtering (Figure 8a). We filter away all other variables, keeping only 11 of the original 18, and obtain the layout in Figure 8b. The colors of the remaining cells change to reflect the range of similarities present in the recomputed dendrogram after filtering.

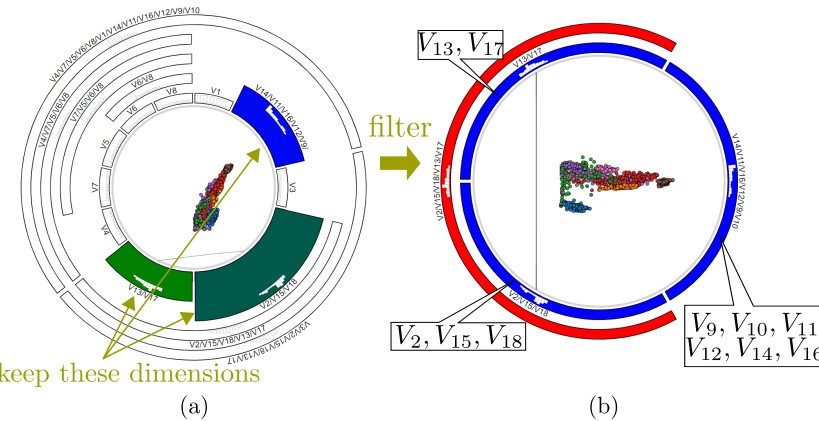

**Figure 8.** (**a**) Variables to filter away (white). (**b**) RadViz++ result after variable filtering (using 11 of the original 18 variables).

### 3.5. Data-to-Data and Data-to-Variable Analysis

As mentioned in Section 1, while RadViz-class methods are desirable when one wants to explore both instances and variables simultaneously, other dimensionality reduction (DR) methods exist. State-of-the-art methods, like LAMP [33] and t-SNE [43], achieve in general a (much) better similar-point cluster segregation, which is an important data-to-data analysis task [2]. However, such methods do not provide ways to explain how *variables* determine such clusters.

We combine the strengths of the RadViz metaphor (seeing both instances and variables, explaining instances by variables) and DR projections (better cluster segregation) by displaying in the inner circle scatterplots created by any such DR methods, instead of the force-based RadViz one. To explain projected groups, we next allow users to smoothly *animate* the DR scatterplot towards the RadViz scatterplot, and vice-versa. This way, one can visually focus on a point group, clearly shown in the DR scatterplot, then see where the group goes in the RadViz scatterplot (following the animation), and finally use the RadViz++ mechanisms to explain the points. Figure 9 shows several frames from the animation between RadViz and LAMP scatterplots for the Segmentation dataset. Using animation to link different displays of the same data, in particular merging insights obtained from different types of DR projections [44], but also for other data types such as 3D data volumes [45], trail sets [46], and 2D images [47], has been proven to be very effective. As demonstrated in all these works, animation is superior to using (two) spatially-distinct views linked by classical brushing-and-selection for the task of relating elements (groups of data points) shown in the two views. The topic is further discussed in [48]. Key to this are the facts that (1) users can focus on a single view in the animation case, rather than having to continuously switch looking at two views; and (2) can spot structures of interest, e.g., forming or splitting groups of points, that appear at *any* moment during the animation, but are not visible in the end views. Moreover, using a single view increases the visual scalability of the method, i.e., allows it to show larger datasets in the same screen space.

Differently from RVD, we do not show a *static* interpolation of two projections (in its case, RadViz and a spring-based system) to explain a better-clustered plot in terms of the anchors, as this may misguide the user, as already discussed in Section 2.2. Therefore, our combination of a DR projection with a RadViz explanation is to our knowledge novel. However, when the LAMP plot is shown (at the end of the animation), users may be confused by the visual presence of the anchors, and aim to interpret the positioning of points in LAMP in terms of the anchors, which would be incorrect. To alleviate this, we add a gray background behind the anchors in the icicle plot. When the background is visible (gray), it tells we are in LAMP mode, so the anchor positions should not be considered (Figure 9 right); when it is invisible (white), it tells we are in RadViz mode, so anchors explain the scatterplot point positions (Figure 9 left). During the animation, the background color linearly changes between its two end colors, indicating that we have a transitional state. An alternative we considered

is to make the anchors transparent in LAMP mode. However, this would not allow us to explain point groups by variables values.

Figure 10a shows the result of LAMP in RadViz++ for the Segmentation dataset. Compared to the RadViz force-based layout (Figure 8 and earlier), we now see a much better cluster separation. Animating this view towards the RadViz layout (Figure 8) allows us to explain these clusters in terms of the data variables, as discussed so far.

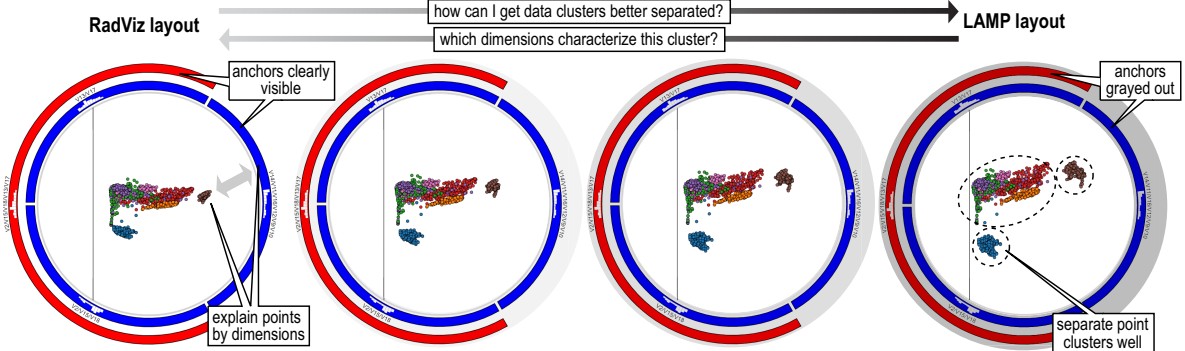

**Figure 9.** Animation of RadViz scatterplot (**left**) towards the LAMP scatterplot (**right**) for the Segmentation dataset. Interpolation factors are $0.2, 0.4, 0.6, 0.8$. While the LAMP plots offer better cluster segregation, the RadViz plot explains the points in terms of variables. Note how the icicle-plot background opacity changes to indicate the RadViz *vs* LAMP mode of the scatterplot.

DR projections can also benefit from variable filtering (Section 3.4.2). Figure 10b shows LAMP applied to the variables selected after the filtering done in Figure 8. We see the same cluster separation as when using LAMP on all 18 variables (Figure 10a). We obtain a DR projection having roughly the same clustering quality as the original one, but with about half (11) of the original 18 variables.

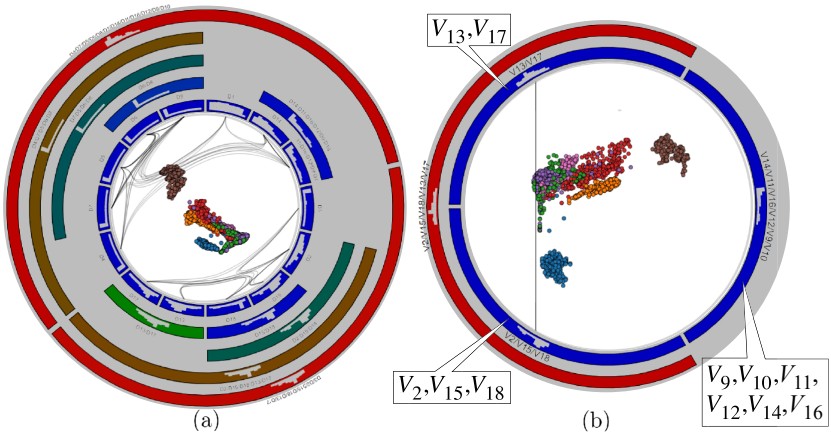

**Figure 10.** (**a**) LAMP scatterplot for the Segmentation dataset. (**b**) LAMP after the variable filtering shown in Figure 8, leading to a better clustering, but using only 11 of the 18 variables.

While force-based point positioning, and also variable-range filtering (Section 3.3), implicitly explain *all* scatterplot points by variables and their ranges, one often wants to explain a *specific* group of points. We support this by a brushing-and-linking tool that links brushed and/or selected points (in the scatterplot) to their histogram bins (in the circular icicle plot) where their values reside. We show the linking by drawing lines between points and bins. To reduce visual clutter, we use again bundling to group these lines. Brushing-and-linking tool is bidirectional, as we can also select bins and show all points having values therein, as shown in Figure 11. We selected here two clusters in the LAMP scatterplot of the Segmentation dataset. For each cluster, bundles show how its points can be explained by specific ranges (bins) of the three variable-sets used in the analysis.

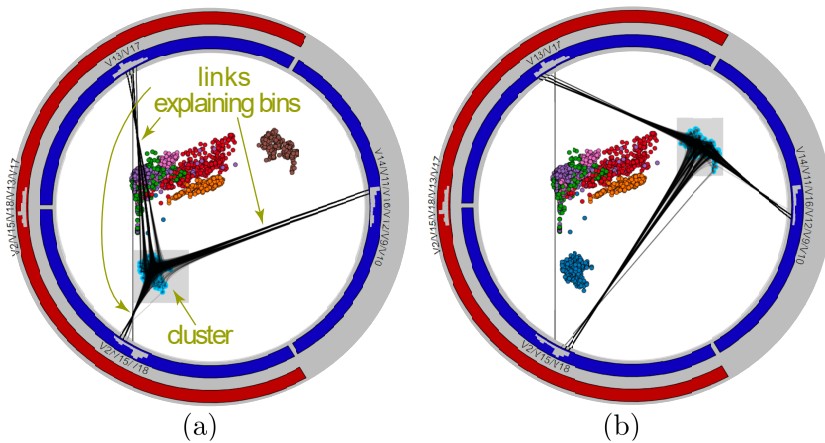

**Figure 11.** Brush-and-link explanation of the (**a**) blue and (**b**) brown clusters. Despite groups of points cannot be correlated to anchors in the LAMP scatterplot, it still valid to explain them in terms of variable ranges.

## 4. Experiments

We next illustrate the working and added-value of RadViz++ with experiments on three different datasets. First, we validate our method using a synthetic dataset, for which ground-truth is known (Section 4.1). Next, we compare RadViz++ with other high-dimensional visualization methods and show that we can reach the same conclusions (Section 4.2). Finally, we present the analysis and obtained insights from a complex dataset (Section 4.3).

### 4.1. Validation on Synthetic Data

We use the dataset described in [5] to validate our method. In their article, the authors proposed several visual metaphors (different from ours) to explain the projected data by their variables. The dataset has $m = 350$ instances, $n = 3$ variables, and $|C| = 2^n - 1$ clusters. Each cluster $c \in C$ contains instances having variation in only a subset of the $n$ variables, while the rest is set to zero. In this sense, clusters $c_1, \ldots, c_7$ contain instances with variation in the variables $\{V_1\}, \{V_2\}, \{V_3\}, \{V_1, V_2\}, \{V_1, V_3\}, \{V_2, V_3\}$, and $\{V_1, V_2, V_3\}$. Data variation in each cluster $c$ follows a different normal distribution $\mathcal{N}(\mu_c, \sigma_c)$ centered at $\mu_c$ and with standard deviation $\sigma_c$. The dataset was created with $(\mu_1, \ldots, \mu_7) = (0, 5, 7, 30, 40, 30, 20)$ and $\sigma_1, \ldots, \sigma_7 = 0.5$.

The authors visualized this dataset using LAMP (Figure 12a). The LAMP projection is binned on a uniform 2D grid based on user settings, where a clustering algorithm takes place. For each found cluster, histograms show the variance of the variables of the contained data points. Briefly put, the method shows clusters in the data and also which variables are (mostly) responsible for their formation.

We next use RadViz++ to find and explain clusters in this dataset (Figure 12b). For visual inspection, we color scatterplot points by their respective cluster IDs. Here and next, these IDs are not used as variables in RadViz++. In the result, we see that the scatterplot contains 7 distinct point clusters $c_1, \ldots, c_7$. The *positions* of these clusters with respect to the 3 variables directly provide the needed explanations, without needing more complex interaction, linked-views, comparing heights of bars in different histograms, or data gridding as in [5]. Equally importantly, the explanations of clusters in terms of variables in our case are the same as those provided by pagliosa et al. [5]. However, in RadViz++ instances whose variation occurs only in one variable, i.e., those in clusters $c_1, c_2, c_3$, are mapped to the same 2D locations, due to limitations of the RadViz force-based placement scheme (see Section 2.1). To decrease this visual ambiguity, we use the brushing-and-linking tool (Figure 13) to select each such point-like cluster in the scatterplot. Since we see edges going from a cluster to *multiple* bins in at least one variable, this explicitly shows that there are *multiple* points mapped to the same scatterplot location. A tooltip could inform details about the selected points for further analysis.

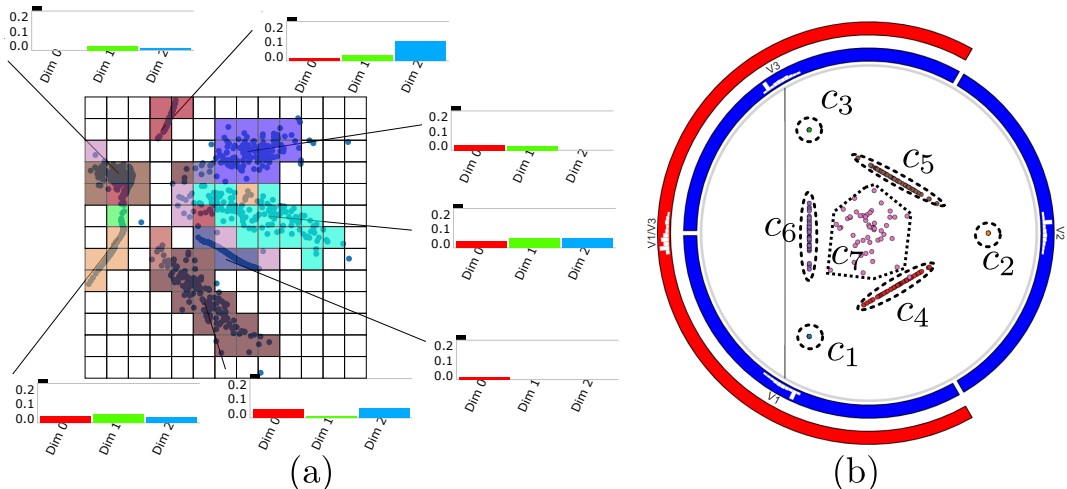

**Figure 12.** (**a**) Attribute-based analysis of 7 Gaussian clusters dataset [5]. The variable 'Dim *i*' maps to $V_{i+1}$ in our notation. (**b**) RadViz++ leads to the same conclusions with a cleaner and simpler layout.

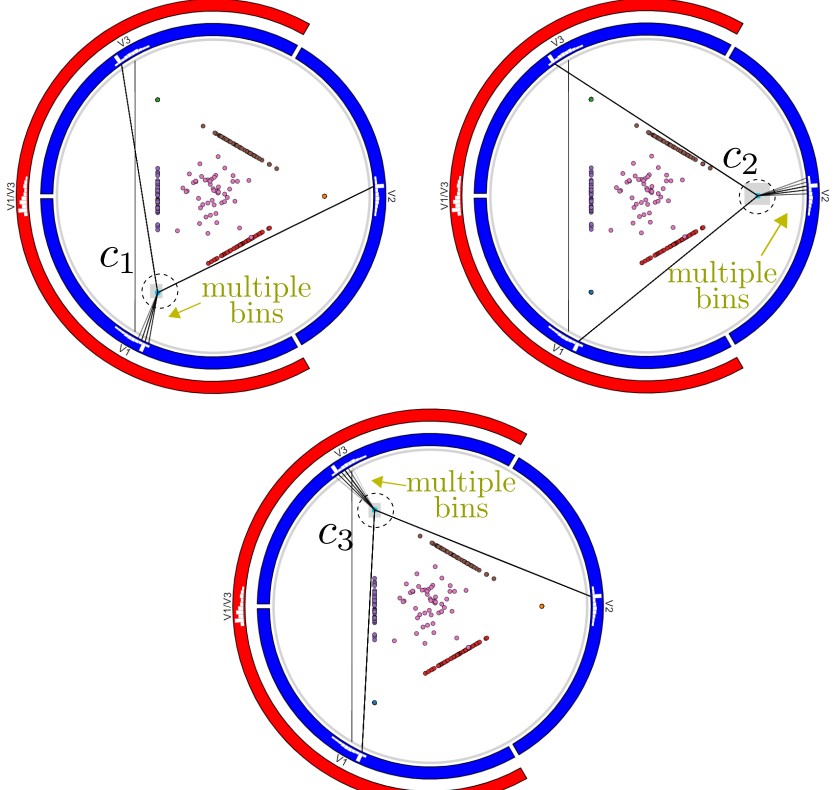

**Figure 13.** The brush-and-link tool helps explain clusters whose points overlap in the scatterplot, thereby decreasing ambiguity problems. For each selected cluster $c_i$, bundles show that its points have *multiple* values in at least one variable bins.

### 4.2. Wisconsin Breast Cancer

This dataset is commonly used as benchmark in visualization and machine learning (see the extensive reference list in [35]). It has $m = 699$ instances (patient tissue samples), $n = 9$ variables (microscopic tissue data), and 2 labels (cancer or lack thereof). The aim is to find which variables or ranges of variables that help to predict the class labels, much as for the Segmentation dataset. We again compare RadViz++ with [5] to verify if we can achieve the same conclusions.

In their article, Pagliosa et al. [5] conclude that both clusters (for the two existing labels) mainly differ because of the *variance* of specific variables. This is shown by the boxplots in Figure 14. The bottom (orange) cluster, corresponding to malignant instances, is described by a high variance in almost all variables. The top cluster (benign instances) has a low variance in all variables except *Clump Thickness*. In addition, one can also conclude that *Mitosis* is the least discriminant variable between the two clusters, as it has quite low variance in both.

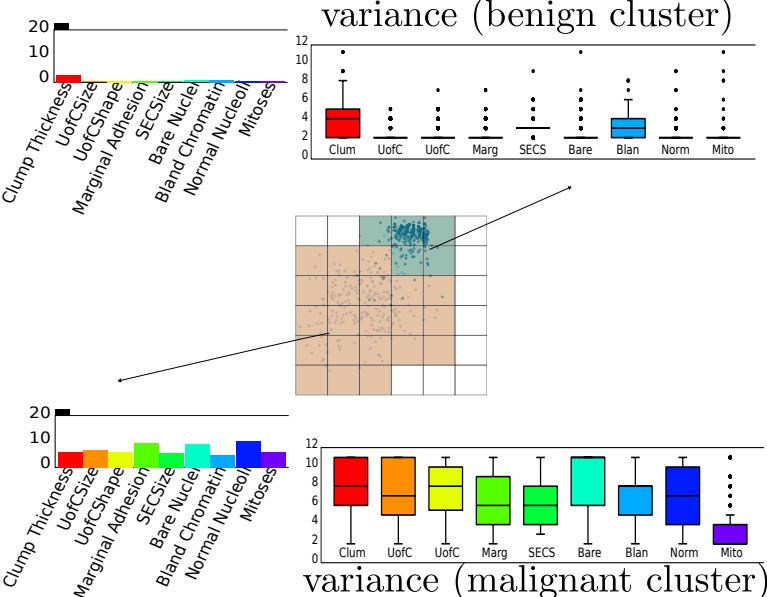

**Figure 14.** Breast Cancer dataset analysis performed by Pagliosa et al. [5]. The variance of the involved variables is the main discriminative factor between the two clusters. All variables contribute quite similarly to discrimination, except *Mitosis*, which has a low overall variance.

We next use RadViz++ for this dataset (Figure 15a). The edge bundles show directly that the *Mitosis* anchor is the only one that has no edge to other anchors, which indicates that that variable has the lowest correlation with all others. As we saw, this is confirmed in Figure 14. Conversely, the most opaque edge connects the *Uniformity of Cell Size* (*UofCSize*) and *Uniformity of Cell Shape* (*UofCShape*) variables. Also, their high-correlation is depicted by their blue-parent node in the icicle plot. Note that the visualization in Figure 14 cannot show this insight. Besides these extremes, Figure 15a shows no other significant clusters or correlation differences. This tells that the remaining variables have similar correlation coefficients. In this case, it is not a good option to analyze this dataset using the force-based scatterplot metaphor as proposed by RadViz, since this will map all instances close to the circle center, as we indeed see in Figure 15a.

To find which variables discriminate between the two clusters, and why, we use the LAMP scatterplot in RadViz++ (Figure 15b. As expected, this scatterplot separates clusters. We now use brushing-and-linking to explain these in terms of variables. We first select points in the benign (blue) cluster (Figure 16a, then in the malignant (orange) cluster (Figure 16b, and compare the two views to find similarities and differences as follows. First, we see that edges from the benign cluster (Figure 16a go to *multiple* bins of the same variable in both cases, except for variable *Mitosis*, where edges go mainly to the lowest-value bin. Hence, *Mitosis* has a much lower variance for benign instances than the other variables (confirmed in Figure 14). Secondly, we see that bundles for the benign cluster (Figure 16a are more concentrated than bundles for the malignant one (Figure 16b. Hence, variables have a higher variance for the latter than the former instances (again, confirmed by the boxplots in Figure 14). Thirdly, we see that bundles go mainly to the low-side bins of their respective histograms in Figure 16a, while bundles in Figure 16b go more uniformly to all bins, and sometimes more to high-side bins in their respective histograms. The figure illustrates this for the *UofCShape* variable,

but the same is visible for most other variables. Hence, benign instances have overall lower variable values than malignant ones. This finding also matches Figure 14.

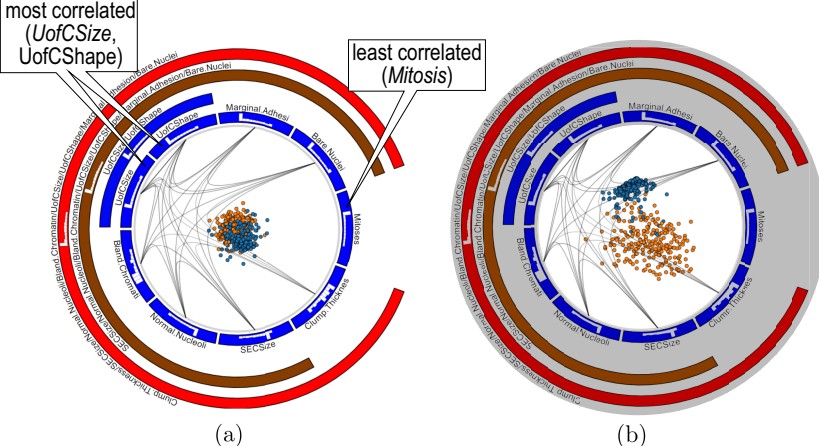

**Figure 15.** Breast Cancer dataset analyzed using RadViz++ with force-based (**a**) and LAMP (**b**) projection.

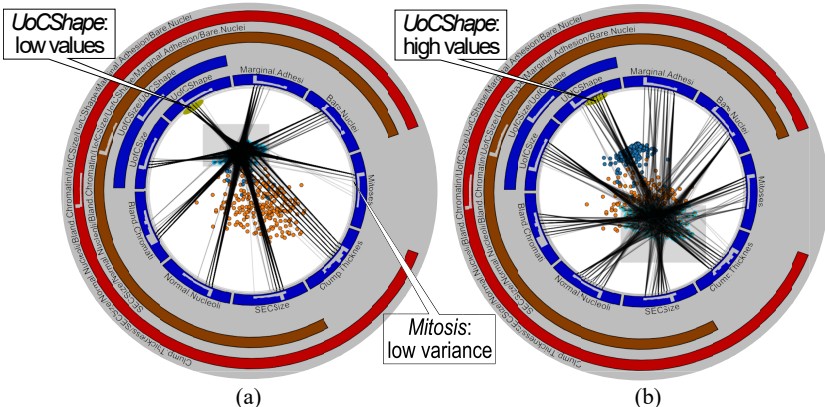

**Figure 16.** Breast Cancer dataset, explaining the benign (**a**) and malignant (**b**) clusters by variables.

We conclude that RadViz++ can lead to the same insights as [5]. However, RadViz++ requires no multiple linked views, data gridding, or other user settings present in the latter, which should make it easier to use. Moreover, RadViz++ allows a *fine grained* linking of variables, and their ranges (bins) to user-specified sets of points in the scatterplot. The technique in [5] cannot do this—it only shows aggregated boxplot statistics for entire classes.

*4.3. Corel Dataset*

Finally, we test our method using the Corel dataset [34], composed of $m = 1000$ images, $n = 150$ SIFT descriptors ($V_1, \ldots, V_{150}$) and 10 class labels. As for the other datasets, visual exploration aims to find correlations of variables (or their properties, such as ranges or variance) with the respective image classes, to further help classifier engineering. This is a much more challenging dataset as the previous ones, not only because of the larger number of classes, but because of its higher dimensionality. In particular, methods such as RadViz, RadViz Deluxe, or the other methods discussed in the related work cannot easily handle 150 variables.

Figure 17 shows the RadViz++ visualization of this dataset, which lets us draw several insights. First, we see that same-class clusters get formed, although not well separated. However, we also see that the 150 variables get partitioned quite clearly into 9 groups, each indicated by a set of mutually bundled edges. This suggests that we could strongly reduce the dimensionality of the data, by

variable aggregation and/or filtering, and thereby possibly achieve a better cluster separation and, thus, explanation.

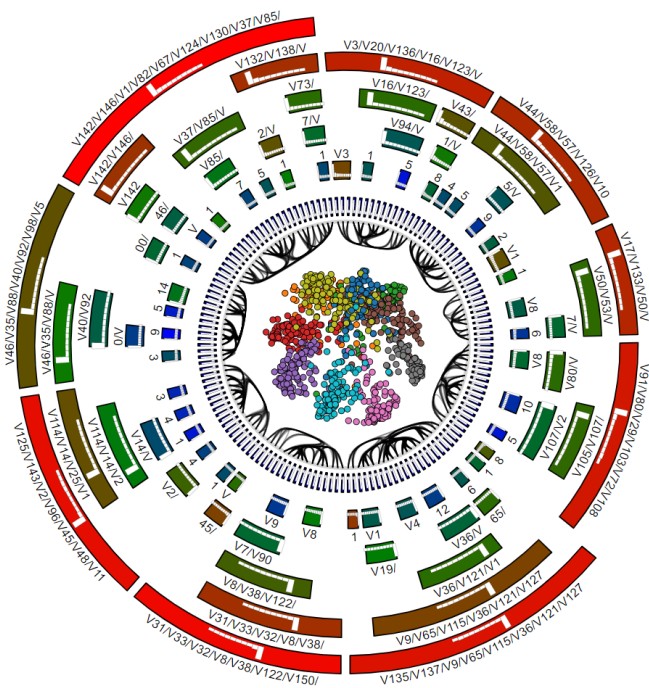

**Figure 17.** Corel dataset visualized using RadViz++.

Following the above observations, we next proceed to aggregate/filter variables. First, we aggregate variable-groups having a medium-range correlation, by selecting their respective nodes, marked green in the icicle plot. Figure 18a shows the start of the selection process, where four such groups are highlighted by the corresponding green-hue nodes in the icicle plot. After a few extra aggregation operations, we obtain the simplification shown in Figure 18b. The ten variable groups present in the figure fairly describe the underlying data, as each variable group describes well one of the 10 classes, seen as 'pulling' the points of the respective class towards its anchor. To see if we can improve class separation, we add a few more variable-groups to the selected ones (yellow '+' signs in Figure 18c). However, this addition does not improve the class separation—compare the scatterplot in this image with the earlier one in Figure 18b. Hence, we revert this step, going back to the variable-groups shown in Figure 18b. Finally, we create a new layout using only the selected variables (Figure 18d). We can see now how each class is strongly 'pulled' towards a single anchor, corresponding to the variable-set that describes it best. Of course, the cluster separation is not perfect—there still is a number of points in the center of the scatterplot, which require most of the selected variables to be described. Finding such points is actually useful, as these are difficult classification cases.

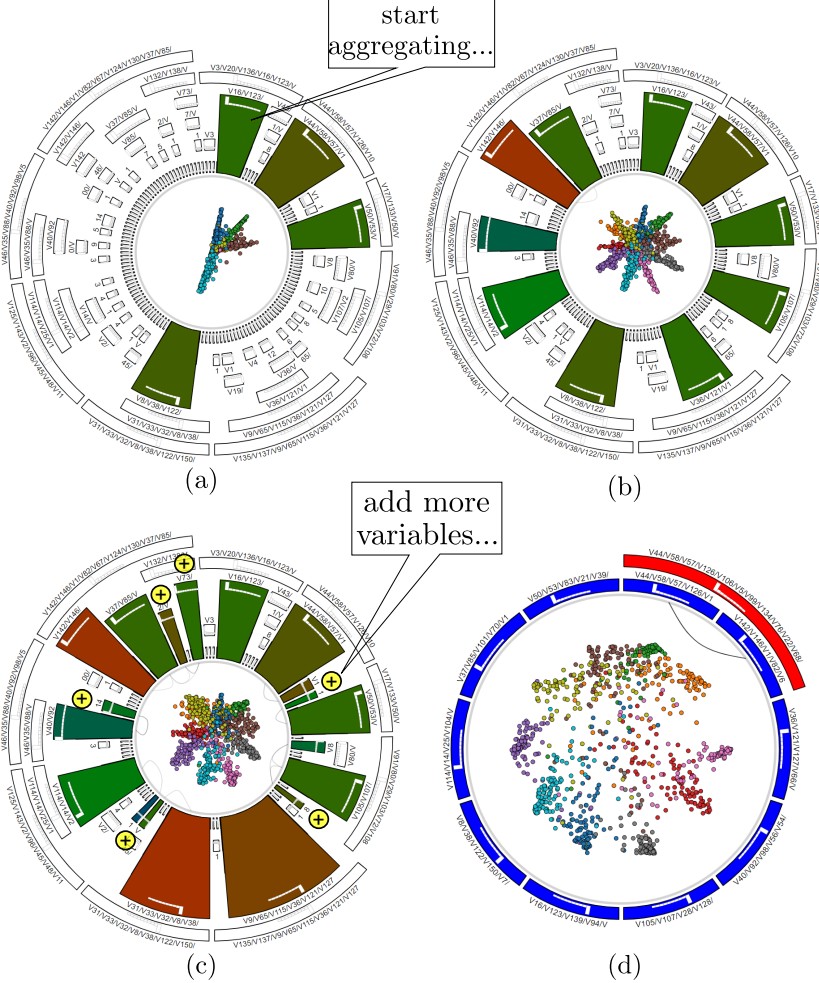

**Figure 18.** Finding the most descriptive variables for the 10 clusters in the Corel dataset. Detailed description in the text.

We can next use interactive variable selection to verify how each of the 10 variable-groups we ended with (Figure 18d) indeed explain the data clusters. For this, we deselect all these variable-groups and next select (activate) them one-by-one. Figure 19a–c show three such selection steps. We can now see quite well how each variable-set is responsible for explaining a separate cluster, as points having the respective cluster color get clearly 'pulled' towards the respective selected anchor. Indeed, if the variable-sets we created would not explain well the data clusters, then activating them would pull points having mixed colors (of many different classes) towards the respective anchors. Finally, we consider further aggregating (simplifying) the variable-set we obtained so far. For this, we aggregate the two variable-sets which are children of the red parent node in the icicle plot in Figure 18d. Figure 19d shows the result. Even if the red color of the parent node had not been a sufficiently strong warning that the respective variable-sets are very dissimilar (uncorrelated), we can see in the scatterplot in Figure 19d that the top-right green and orange clusters, which were quite well separated before aggregation (Figure 18d), now get mixed up under the aggregated set of variables. Hence, we revert this aggregation and end the exploration with the 10 variable-sets shown in Figure 19d as being the best ones for explaining the 10 clusters in the dataset.

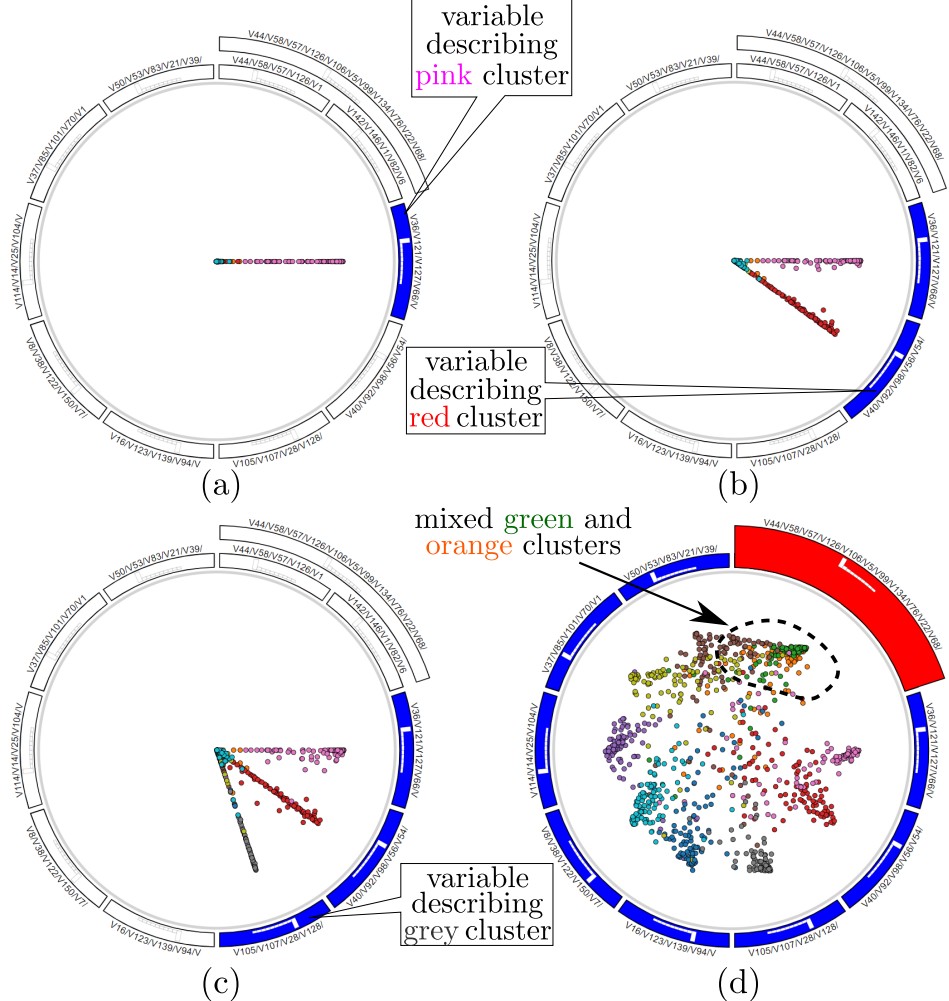

**Figure 19.** Verifying the explanatory power of each variable-set after selecting its respective anchor (**a**–**c**). Further aggregating these variables reduces cluster separation (**d**), so should be avoided.

## 5. Discussion

We next discuss our proposal vs. the requirements R1–R4 (Section 1):

**R1:** Our method is as scalable as all other scatterplot-based visualization techniques in the number of *instances*, as every one is mapped to a 2D point. *Variable*-wise, we argue that our method scales far better than all existing RadViz-class techniques due to the hierarchical variable aggregation and variable filtering. Two aspects are related to this point, as follows. First, even when hierarchical variable aggregation is not used, we can display up to roughly thousand variables along the plot circumference, since each variable requires only a circle sector of a few pixels width to be visible and distinct from its neighbors. This same scalability has been demonstrated earlier by visual designs using the same radial icicle plot, see e.g., [13,49,50] for applications visualizing thousands of elements from software hierarchies. Secondly, as explained in Section 3.2.1, we simplify the hierarchy produced by agglomerative clustering based on a user-defined similarity factor $\delta$ (preset to 10% of the root cluster diameter). As explained there, this factor controls the number of levels the simplified hierarchy will show, so users can 'flatten' arbitrarily large hierarchies in this way up to the desired level of detail. Also, it is important to note that we do not need to display, nor even compute, the *full* variable hierarchy: If, during the bottom-up clustering process, we decide that we reach a point where the dissimilarity of variable-groups (roots of hierarchy subtrees computed so far) is larger than what the user can tolerate, then we can simply stop clustering and only use the hierarchy levels computed so

far. This explicitly limits the maximum number of levels (concentric rings in the icicle plot) that will be present in RadViz++. Finally, users can always *locally* refine the level-of-detail by choosing to aggregate certain groups of variables (hierarchy subtrees) but show other ones in full detail. The same techniques have been successfully used to visualize hierarchies of tens of levels and thousands of leaf nodes, as mentioned earlier [49,50]. In the same time, the hierarchy allows a flexible variable-placement along the RadViz circle where similar variables are placed close to each other.

**R2:** We decrease ambiguities of data-to-variable analyses by histogram bins and brushing-and-linking that shows which variables (and their ranges) correspond to a user-specified given subset (cluster) of scatterplot points. Separately, we decrease such ambiguities by variable filtering and aggregation, which allocates more visual space to explain fewer variables—thus, more space per variable. We also bundle bin-to-cluster links to further decrease visual clutter and associate data points to variable ranges (bins) easier.

**R3:** Besides the aforementioned hierarchy-based anchor placement and dendrogram of clustered variables, we use hierarchical edge bundles (HEB) to explicitly show groups of similar variables. HEB is spatially compact, intuitive, and also explains anchor-placement ambiguities which are inherent to the RadViz circular layout.

**R4:** To better separate point clusters, we allow exploring data by two different dimensionality-reduction methods. At one extreme, the RadViz projection explains well instances in terms of variables, but may not separate point clusters well. At the other extreme, the LAMP projection achieves the opposite. Users can fuse insights provided by the two projections by e.g., selecting clusters of interest (in LAMP) and animate them back-and-forth to the RadViz projection, which explains them in terms of variables (or conversely).

*Limitations*

While scalable, simple to implement, and working generically for any quantitative high-dimensional dataset, our proposal also has several limitations. First, even when doing variable aggregation and filtering, a certain amount of visual overlap of different-value instances will occur into the scatterplot, due to inherent limitations of the RadViz placement (Equation (3)). While other placement methods may improve upon this, e.g., RadViz Deluxe [14], we chose to do this via a radically different way, namely using a different DR method (LAMP) and animation to link it with the anchor placement. Whether our approach is better than RadViz Deluxe in terms of ease of use and accuracy of the obtained insights is an open problem requiring further evaluations. Secondly, our approach cannot yet handle categorical data; also, handling negative data values is subject to limitations present, to our knowledge, in all other RadViz-class methods. Extending our hierarchical anchor placement based e.g., on similarity metrics defined on categorical data [51] is an interesting possibility yet to be explored.

## 6. Conclusions

We have presented RadViz++, a set of techniques for interactive exploration of high-dimensional data using a RadViz-type metaphor. We designed our techniques to alleviate several types of problems present in existing RadViz-class methods, as follows. We increase variable scalability by using a variable clustering technique and simplified variable-hierarchy visualization, which allows us to easily handle over a hundred variables. We reduce ambiguities of the RadViz circular layout, and also summarize variable similarities by using a hierarchical edge bundling approach. We explain data clusters in terms of variables and variable-ranges by linking the former with histogram bins representing the latter. Finally, we reduce visual clutter to better analyze data clusters by integrating a separate

dimensionality-reduction method, good at cluster segregation, and linking its explanation with the RadViz metaphor via animation. We show that our approach can lead to the same insights on two different datasets as when using existing visualization methods, but with less effort, and demonstrate scalability on a third dataset.

Several future work directions are possible. First, extending RadViz++ to handle effectively negative data values and also categorical data would increase the application range considerably. Secondly, studying different variable similarity metrics apart from correlation could lead to approaches that can explain more effectively more complex patterns in high-dimensional data. Finally, experimental studies comparing RadViz++ to other high-dimensional RadViz-class visualizations for a wide benchmark of datasets and tasks will lead to insights allowing the specific fine-tuning of our method for increased specificity and effectiveness.

**Author Contributions:** Conceptualization, A.C.T.; methodology, L.d.C.P. and A.C.T.; software, L.d.C.P.; validation, L.d.C.P.; formal analysis, L.d.C.P.; data curation, L.d.C.P.; writing—original draft preparation, L.d.C.P.; writing—review and editing, L.d.C.P. and A.C.T.; visualization, L.d.C.P.; supervision, A.C.T.; project administration, A.C.T.

**Funding:** This research was funded by FAPESP (São Paulo Research Foundation), Brazil, under grant 2018/10652-9.

**Acknowledgments:** We acknowledge the sponsorship of FAPESP.

**Conflicts of Interest:** The authors declare no conflict of interest. The funders had no role in the design of the study; in the collection, analyses, or interpretation of data; in the writing of the manuscript, or in the decision to publish the results.

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
