# Peer review of "RadViz++: Improvements on Radial-Based Visualizations"

_informatics, doi:10.3390/informatics6020016_

Reviewer 1 Report

The paper has presented a RadViz extension for interactive exploration of high-dimensional data.

An implemented interactive tool can strengthen the significance of the research.

Some typing errors:

- tree -> three (Page 1).

- wherei nstances -> where instances (Page 4)

- (v) -> (b) (Caption of Fig. 1)

Author Response

Dear Reviewer, 

We thank you for your kind support and for the useful comments and observations about our manuscript. We carefully proofread the paper and corrected the errors according to the received suggestions.

Reviewer 2 Report

This paper makes an important contribution to address some of the problems of RadViz plots, such as, scalability in the number of variables and ambiguities in the data. The problem of desimbaguation and scalability has been addressed in related multidimensional visualizations such as SC. For instance, data projections of the points have been modified to avoid ambiguities (Adaptable radial axes plots for improved multivariate data visualization, M Rubio‐Sánchez, A Sanchez, DJ Lehmann. Computer Graphics Forum 36 (3), 389-399, 2017) and variables have been grouped into "supervariables" in order to reach scalability (iStar: An interactive star coordinates approach for high-dimensional data exploration, G. Garcia Zanabria, LG Nonato, E Gomez Nieto. Computers & Graphics 60: 107-118, 2016). Due to the relationship between RadViz and SC it would be good to comment on both points and how they are addressed in both alternatives.

Specifically, the authors propose RadViz++, a set of techniques for interactive exploration of high-dimensional data using a RadViz-type metaphor. The idea is original, there's work behind it, and it can have some applications as demonstrated in the paper. Nevertheless, some of them can be a problem for data exploration, since users could force what they want to see instead to analyze the real data structure. For instance, the inclusion of the LAMP scatterplot can offers better cluster segregation, but its representation in the same icicle plot than in RadViz can confuse the analyst. Domain analysts may have the feeling that variables pull on the data points following RadViz method when the projection is non-linear and they cannot get information from the placement of the variables. It may be more interesting to set aside the LMAP plot and enable linking & brushing with the RadViz plot. Additionally, scalability with respect to the number of variables is not fully justified. The space occupied by the icicle plot is very high in relation to the display of the data points. If a high number of variables is included, there would not be enough space to show them and see the hierarchical relations (for instance, scenarios with thousands of variables like genes). This should be better explained.

Some other shortcomings include that the paper is not perfectly aligned with the journal (the authors should place more emphasis on human-computer interaction rather than on information visualization). In this sense, a user study should be advisable to improve this part of the paper. In addition, it is advisable to check some small flaws, like data-do-data, ...

Author Response

Dear Reviewer,

We thank you for your kind support and for the useful comments and observations about our manuscript. In order to fill previous gaps and improve the quality of our article, we have modified the text according to the received suggestions, as reported the PDF file attached. 

Round  2

Reviewer 2 Report

The topic of this manuscript is interesting. Organization of the paper is good and the proposed method is quite novel. The authors have addressed my comments quite well. As a conclusion, it is a very nice research.

Author Response

Dear Reviewer,

We are pleased to know that our manuscript has now met your expectations. We thank you for the good comments and suggestions.

Sincerely,

Lucas Pagliosa and Alexandru Telea.